# The effect of adding neuromuscular electrical stimulation to exercise therapy on patellofemoral pain: A systematic review and meta-analysis

Jiawei Zheng[1], Zixuan Wei[1], Huiwu Zuo[2], Kunpeng Wang[1], Xikai Lin[3], Jian Chen[1]*

1 Engineering Research Center of Sports Health Intelligent Equipment of Hubei Province, Wuhan Sports University, Wuhan, China, 2 School of Exercise and Health, Shanghai University of Sport, Shanghai, China, 3 Beijing University of Chinese Medicine Shenzhen Hospital (Longgang), Guangdong, China

* 2007021@whsu.edu.cn

## Abstract

### Background

This study investigated the effects of adding neuromuscular electrical stimulation (NMES) to exercise therapy on pain, knee function, quadriceps strength, and the ratio of activation of the Vastus Medialis Oblique (VMO) to Vastus Lateralis (VL) muscles in people with Patellofemoral pain (PFP).

### Methods

A rigorous search for randomized controlled trials (RCTs) spanning database inception to July 1, 2024, was executed across PubMed, Embase, Web of Science, Cochrane Library, and Scopus. Two researchers independently screened the literature and extracted data. The Cochrane Risk of Bias Tool was used for included studies to assess risk of bias and the GRADE system was used to assess the certainty of evidence for outcomes.

### Result

Nine randomized controlled trials, encompassing 337 participants (171 intervention, 166 control), were included in the analysis. The meta-analysis findings indicated that compared to exercise therapy alone, NMES combined with exercise therapy significantly reduces pain intensity (MD: −0.37; 95% CI: −0.64 to −0.10; P = 0.007), notable improvements in knee function (MD: 4.76; 95% CI: 2.08 to 6.84; P = 0.0002), and significantly increased quadriceps muscle strength (SMD: 0.55; 95% CI: 0.24 to 0.87; P = 0.0006). However, there was no significant impact observed on the VMO/VL ratio (SMD: 0.8; 95% CI: −0.33 to 1.93; P = 0.16). Subgroup analyses superimposed that incorporating NMES did not result in a meaningful reduction in pain intensity (MD: −0.85; 95% CI: −1.76 to 0.07; P = 0.07) or a significant improvement in quadriceps

**Data availability statement:** All relevant data are within the paper and its Supporting Information files.

**Funding:** The author(s) received no specific funding for this work.

**Competing interests:** The authors have declared that no competing interests exist.

muscle strength (SMD: 0.27; 95% CI: −0.24 to 0.78; P = 0.30) when the treatment period was less than or equal to 4 weeks.

## Conclusion

The study's findings revealed that adding NMES to exercise therapy offers further improvement in pain intensity, knee function, and quadriceps strength in people with PFP compared with exercise therapy alone, but did not significantly improve VMO/VL ratio, and it recommended that the duration of the intervention lasts more than4 weeks for better therapeutic outcomes.

## 1. Introduction

Patellofemoral pain (PFP) is characterized by discomfort or aching around or behind the kneecap, especially during activities such as knee bending, squatting, or running [1]. Prospective research has identified that patellofemoral pain etiology can be categorized into proximal, local, and distal factors [2]. Proximal factors involve hip muscle weakness leading to altered lower limb kinematics during weight-bearing activities [3]. Local factors include quadriceps dysfunction [4], altered Vastus Medialis Oblique (VMO)/ Vastus Lateralis (VL) activation timing [5], quadriceps and hamstring tightness [6]. Distal factors encompass foot and ankle mechanics, such as increased navicular drop, which affect tibial rotation and subsequently alter patellofemoral joint mechanics [7]. All these factors contribute to PFP by creating abnormal patellofemoral joint mechanics and increased joint stress.

The primary treatment for PFP is conservative, focusing on rehabilitation through exercise therapy, which is widely recognized as the main treatment for musculoskeletal conditions including PFP and is often combined with adjunct treatments to achieve optimal clinical outcomes [8,9]. Neural muscular electrical stimulation (NMES), as a painless and non-invasive technique, is typically used as an adjunct treatment method to further promote recovery [10,11]. NMES works by stimulating motor neurons, leading to muscle contractions that strengthen muscles [12,13]. However, whether NMES combined with exercise therapy can produce better clinical benefits for PFP patients currently has no definitive conclusion.

A recent systematic review indicated that NMES combined with exercise therapy can slightly improve pain in patients with PFP, but does not significantly improve knee function. However, the evidence certainty of these results is very low [14]. Furthermore, the review did not evaluate key indicators such as muscle strength and lacked subgroup analysis based on intervention duration. Given the emergence of more evidence and the weakness of existing evidence, further systematic reviews are necessary to clarify the clinical value of NMES as an adjunct treatment for patients with PFP.

This systematic review aims to evaluate the effectiveness of adding NMES to exercise therapy for treating PFP. We compared the outcomes of combined exercise

therapy and NMES with exercise therapy alone. The effectiveness was assessed using four key outcome measures: pain intensity, quadriceps muscle strength, knee function, and the VMO/VL activation ratio.

## 2. Methods

This systematic review follows the Methodological Expectations of Cochrane Intervention Review standards [15], Preferred Reporting Items for Systematic Reviews and Meta-Analyses (PRISMA) [16]. The protocol was formally documented in the PROSPERO database under the registration number CRD42024574963.

### 2.1. Search strategy

Two distinct evaluators (ZJW and WZX) conducted independent searches across multiple databases—PubMed, Embase, Web of Science, Cochrane Library, and Scopus—for randomized controlled trials (RCTs) from the inception of these databases until 01 July 2024. The methodologies for these searches are detailed in Appendix 1.

### 2.2. Eligibility criteria

**2.2.1. Type of study.** This review only included randomized controlled trials published in peer-reviewed journals, with no restrictions regarding language or publication date.

**2.2.2. Types of participants.** We included studies of adults with a diagnosis of PFP according to international consensus statements [1]. We excluded studies of people in the adolescent population and studies with other pathologies such as knee osteoarthritis and patellar fracture.

**2.2.3. Types of interventions.** In the study, the experimental group received exercise therapy combined with NMES, while the control group only received exercise therapy or exercise therapy combined with sham NMES treatment.

**2.2.4. Types of outcome measures.** The metrics for assessing the results of the study comprised evaluations of pain intensity, knee functionality, strength of the quadriceps, and the ratio between VMO and VL. Pain intensity was assessed using both the Visual Analog Scale (VAS) and the Numerical Pain Scale (NPS). The functionality of the knee was evaluated through the Anterior Knee Pain Scale (AKPS). The strength of the quadriceps muscle was determined through isometric or isokinetic strength assessment, while the VMO/VL ratio was quantified using surface EMG technology.

### 2.3. Study selection and data extraction

The search studies were initially imported into Endnote X9. After the software automatically removed duplicates, two investigators (ZJW, WZX) independently assessed the studies using their titles and abstracts before reviewing the full texts of those that remained to eliminate any that failed to meet the inclusion standards. Disagreements were settled through consultation with another researcher (WKP). Two investigators (ZJW, WZX) independently gathered information from the included studies, utilizing a uniform template for data extraction and resolving differences via dialogue. The data retrieved encompass these details: (1) Name of the leading author; the publication year; and the country of the research. (2) The sample size; the age range of participants; and the gender distribution. (3) Details about the intervention program, including its duration; the frequency, duration, and intensity of NMES sessions. (4) The outcome data collected and the timing of measurements.

### 2.4. Risk of bias assessment

The revised Cochrane Risk of Bias tool for randomized trials was used by two independent researchers (ZJW and WZX) to assess the risk of bias of the included studies [17]. This tool evaluates potential bias in randomized controlled trials by examining key areas: bias from the randomization process, deviations from intended interventions, missing outcome data, outcome measurement bias, and selective outcome reporting. Each domain was rated as having a low, moderate, or high risk of bias, ensuring a comprehensive assessment of the studies' internal validity.

## 2.5. Certainty of evidence

The assessors (ZJW, WZX) appraised the evidence's reliability for the outcomes by employing the Grading of Recommendations, Assessment, Development and Evaluations(GRADE) approach. This approach takes into account five criteria (risk of bias, inconsistency, indirectness, imprecision, and publication bias) to ascertain the reliability of the evidence, categorizing it into four levels (high, moderate, low, very low) [18].

## 3. Statistical analysis

Analyses in this study were conducted using Review Manager (version 5.3.5). Only continuous variables were included in the data analysis. The findings were reported as the mean difference (MD) or the standardized mean difference (SMD), each accompanied by 95% confidence intervals. The choice between MD and SMD depended on whether the effect indicators had the same or different units or measurements [19]. Heterogeneity among the studies was assessed using the Q test and $I^2$ statistics. Moderate, substantial, and considerable statistical heterogeneity were defined by $I^2$ values of 30%, 50%, and 75%, respectively [15]. Random effects model was used for analysis due to differences in intervention methods between studies and the limited number of studies [20]. If the heterogeneity was very large, we performed sensitivity analysis to assess whether the meta-analysis results were robust. In addition, we performed subgroup and meta regression to find sources of heterogeneity. Since the studies incorporated showed considerable variation in intervention duration, subgroup analyses were conducted based on the specific intervention times reported in each study, with ≤4 weeks defined as short-term and 4–12 weeks defined as medium-term [21]. Publication bias detection using egger's test for analyses with ≥ 9 included studies, which judges publication bias by the intercept of standard normal regression, and a significant intercept ($p < 0.05$, i.e., statistically significant) indicates publication bias [22].

## 4. Results

### 4.1. Study identification

We searched five English-language databases, retrieving 290 studies. After removing 222 duplicates, 39 studies were excluded based on title and abstract review, leaving 29 studies for full-text screening. Two studies were excluded due to the inability to obtain valid data and the lack of effective responses after contacting the authors. A total of 9 studies were ultimately included, all of which underwent quantitative analysis. The PRISMA flow chart is shown in Fig 1.

### 4.2. Study characteristics

Nine randomized controlled trials with 337 participants were included (171 in the experimental group, 166 in the control group) [23–31]. These trials were conducted in China [30,31], the United States [27,28], Turkey [24,26], India [25,29], and Austria [23]. Intervention durations ranged from 4 weeks to 12 weeks [24–28,30,31]. Outcome measurements varied: five studies measured at the end of the intervention [25,27,29–31], two every 3 weeks until the end [24,28], one immediately post-intervention with a 12-week follow-up [26], and one with outcomes measured one month post-intervention and a one-year follow-up [23]. Subgroup analyses accounted for the varying intervention durations.

The NMES frequency ranged from 20–75 Hz, with six studies using pulse durations of 65–400 μs [24,26,27,29–31], and three did not specify pulse durations [23,25,28]. Eight studies used NMES for 15–30 minutes per treatment, three to five times per week [23–29,31], while one study reported 3 weekly sessions without specifying duration [30]. Six studies used the maximum tolerated intensity of NMES [23,24,27,29–31], but three did not report intensity [25,26,28]. Table 1 displays the characteristics of the intervention.

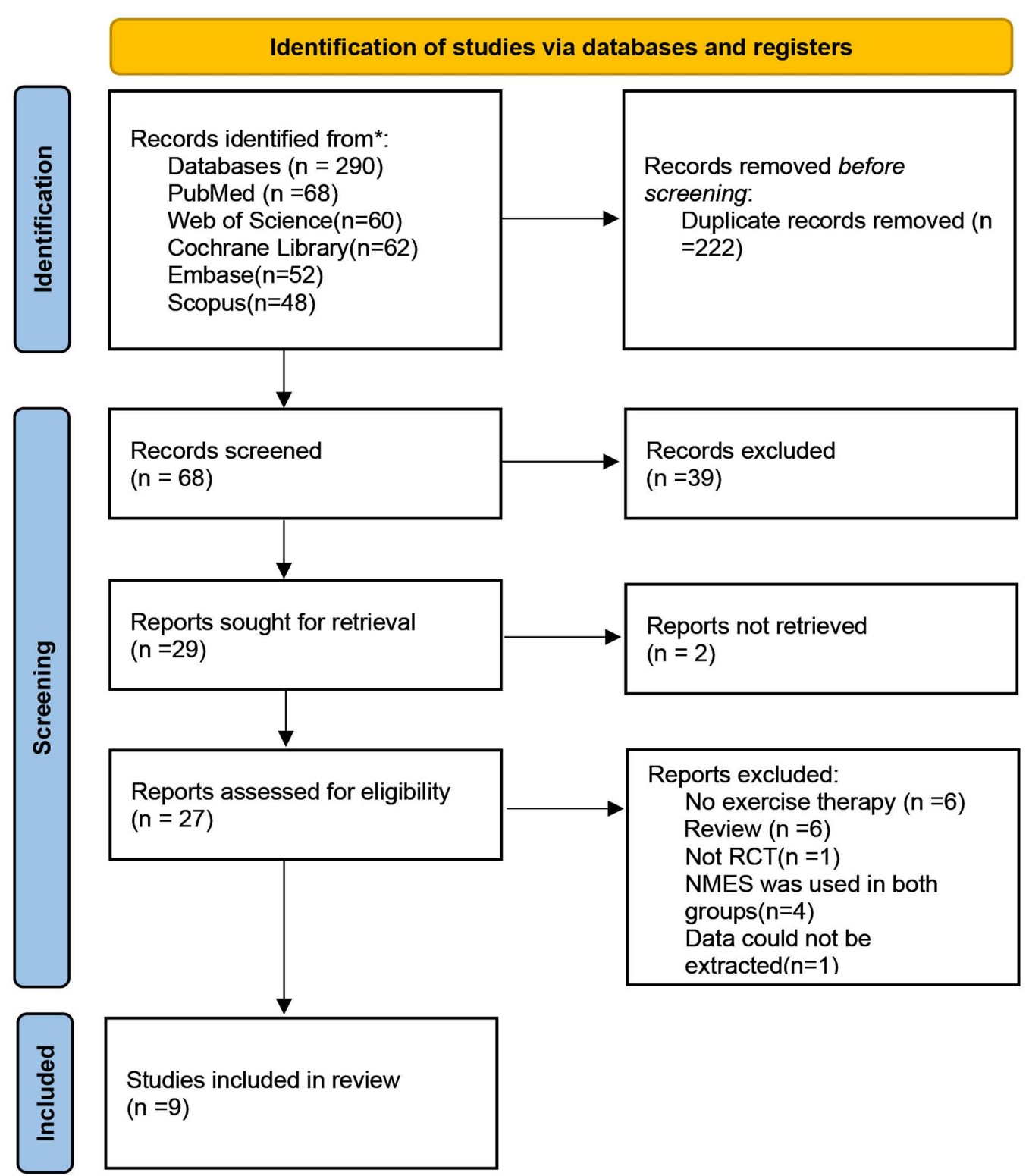

**Fig 1. Flow diagram of included studies.**

**Table 1. Characteristics of the included studies.**

| Study | Country | People(IG:CG) female proportion (IG: CG) | Age(IG:CG) | Experimental group: | Control group: | Intervention frequency, duration, and time | Intervention intensity | Outcome measures |
|---|---|---|---|---|---|---|---|---|
| Akarcali et al., 2013 | Turkey | 20(IG):22(C)G 16/20(IG):15/22(CG) | 41.6±9.58(IG): 36.3±9.59(CG) | NMES+ standardized rehabilitation program | Standardized rehabilitation program | NMES (60 Hz,65–75μs) of VMO for10 min, 5 times per week for 6 weeks | Maximum tolerable intensity | VAS,quadriceps strengths Time points:0,3,6 week |
| Bily et al., 2008 | Austrian | 19(IG):19(CG) 10/19(IG):14/19(CG) | 27±7.7(IG): 23.7±5.5(CG) | NMES+ Physiotherapy | Physiotherapy | NMES(40 Hz) of quadriceps for 20 min,3 times per week for 12 weeks | Maximum tolerable intensity | VAS, AKPS, quadriceps strength Time points:0,1,12 week |
| Celik et al., 2019 | Turkey | 14(IG):13(CG) 9/14(IG):6/13(CG) | 39.1±9.1(IG): 41.5±12.7(CG) | NMES+ standardized rehabilitation program | Standardized rehabilitation program | NMES (50 Hz,400μs) of quadriceps for 20 min,3 times per week for 6 weeks | Not mentioned | AKPS, Lysholm, quadriceps strength, hamstring strength Time points:0,6,12 week |
| Das et al., 2016 | India | 15(IG):15(CG) 11/15(IG):9/15(CG) | 32.87±6.92(IG): 32.67±9.69(CG) | NMES+ exercise program | Exercise program | NMES (30–75 Hz) of quadriceps for 30 min,3 times per week for 4weeks | Not mentioned | VAS, AKPS, VMO/VL Ratio, MVIC of quadriceps Time points:0,4 week |
| Glaviano et al., 2020 | USA | 8(IG):8(CG) 8/8(IG):8/8(CG) | 23.0±6.0(IG): 23.5±4.0(CG) | NMES+ standardized rehabilitation program | Sham therapy +standardized rehabilitation program | NMES (50 Hz,200μs) of quadriceps and GMed for15 min,3 times per week for 4weeks | Maximum tolerable intensity | VAS, Lower-Extremity Kinematics during SLS and SDT tasks, EMG activity of specific muscles during SLS and SDT tasks Time ponts:0,4 week |
| Talbot et al., 2020 | USA | 33(IG):34(CG) 8/33(IG):8/34(CG) | 26.5±3.1(IG): 26.8±6.6(CG) | NMES+HEP | HEP | NMES(50 Hz)of quadriceps for 20 min,3 times per week for 9 weeks | Not mentioned | VAS, quadriceps strength , hamstring strength30-SCST, SCT,6-MWT, SDT Time points:0,3,6,9 week |
| Kumar et al., 2016 | India | 15(IG):15(CG) 11/15(IG):9/15(CG) | 32.87±6.92(IG): 32.67±9.69(CG) | NMES+ exercise program | Exercise program | NMES (30–75 Hz) of quadriceps for 30 min,3 times per week for 4weeks | Not mentioned | VAS, AKPS, VMO/VL Ratio, MVIC of quadriceps Time points:0,4 week |
| Nie et al., 2023 | China | 13(IG):11(CG) 5/13(IG):6/11(CG) | 32.83±8.55(IG): 36.33±6.58(CG) | NMES+ routine functional training | Routine functional training | NMES (20 Hz,200μs,10-20mA) of VMO and GMAX for 20 min,3 times per week for 6 weeks | Maximum tolerable intensity | VAS, AKPS, RMS and IEMG of VMO and VL, VMO/VL Ratio Time points:0.6 week |
| Wu et al., 2024 | China | 18(IG):14(CG) 4/18(IG):5/14(CG) | 20.9±2.7(IG): 21.6±3.8(CG) | NMES+ plyometric training | Plyometric training | NMES (50 Hz,400μs) of VMO 3 times per week for 4weeks | Maximum tolerable intensity | VAS, AKPS, Kinematic and kinetic data of the lower extremity during running Time points:0,4 week |

AKPS, anterior knee pain scale; CG, Control group; CSA, Cross Sectional Area; EMG, Electromyography; FL, Fascicle Length; GMAX, gluteus maximus; GMed, gluteus medius; HEP, home exercise program; IEMG, Integrated Electromyography; IG, Intervention group; Lysholm, Lysholm Knee Scoring Scale; MVIC, Maximum Voluntary Isometric Contraction; NMES, neuromuscular electrical stimulation; PA, Pennation Angle; Q angle, quadriceps angle; QS, Quadriceps muscles strengthened; RCT, randomized controlled trial; RMS, Root Mean Square; ROM, range of motion; SCT, Timed Stair Climb Test; SDT, Step-down task; SLS, Single-leg squat; VAS, visual analogue scale; VL, Vastus Lateralis; VM, vastus medialis; VMO, Vastus Medialis Oblique; 30-SCST, 30-Second Chair Stand Test; 6-MWT, Six-Minute Timed Walk Test.

## 4.3. Risk of bias

Figs 2 and 3 show the risk of bias for each study. We assessed four studies as low risk [25,27,29,31], two studies as having some concerns [23,26], and the other three studies as high risk [24,28,30]. The bias was primarily due to the implementation of blinding methods and incomplete outcome data.

## 4.4. Effect on pain

Eight studies assessed pain using the VAS scale [23–25,27–31]. There was low-certainty evidence and low statistical heterogeneity($I^2$=26%) suggesting that NMES combined with exercise significantly reduced pain compared to controls (MD: −0.37; 95% CI: −0.64 to −0.10; P=0.007) (Fig 4). In subgroup analyses, very low-certainty evidence and considerable statistical heterogeneity ($I^2$=81%) suggested no significant effect when the intervention was less than or equal to 4 weeks (MD: −0.85; 95% CI: −1.76 to 0.07; P=0.07) (Fig 5). However, when the intervention period lasted 4−12 weeks, combined NMES showed significant improvement in pain compared to exercise alone (MD:-0.28; 95% CI:-0.54 to 0.02; P=0.03, very low-certainty evidence), with low statistical heterogeneity ($I^2$=0%) (Fig 5).

## 4.5. Effect on knee function

Six studies measured knee function using the AKPS [23,25,26,29–31]. The meta-analysis showed a significant improvement in knee function with NMES (MD:4.46; 95% CI:2.08 to 6.84; p=0.0002; low certainty of evidence) with moderate statistical heterogeneity ($I^2$=42%) compared to exercise only (Fig 6). For 4–12 weeks interventions, NMES had a significant effect on knee function (MD: 4.19, 95% CI:1.31 to 7.07, P=0.006; very low certainty of evidence) compared to exercise only, though with substantial statistical heterogeneity ($I^2$=52%) (Fig 7).

## 4.6. Effect on quadriceps strength

Five studies assessed quadriceps strength [23–26,28]. NMES significantly improved quadriceps strength (SMD: 0.55; 95% CI: 0.24 to 0.87; P=0.0006; low-certainty evidence) with low statistical heterogeneity ($I^2$=0%) compared to exercise

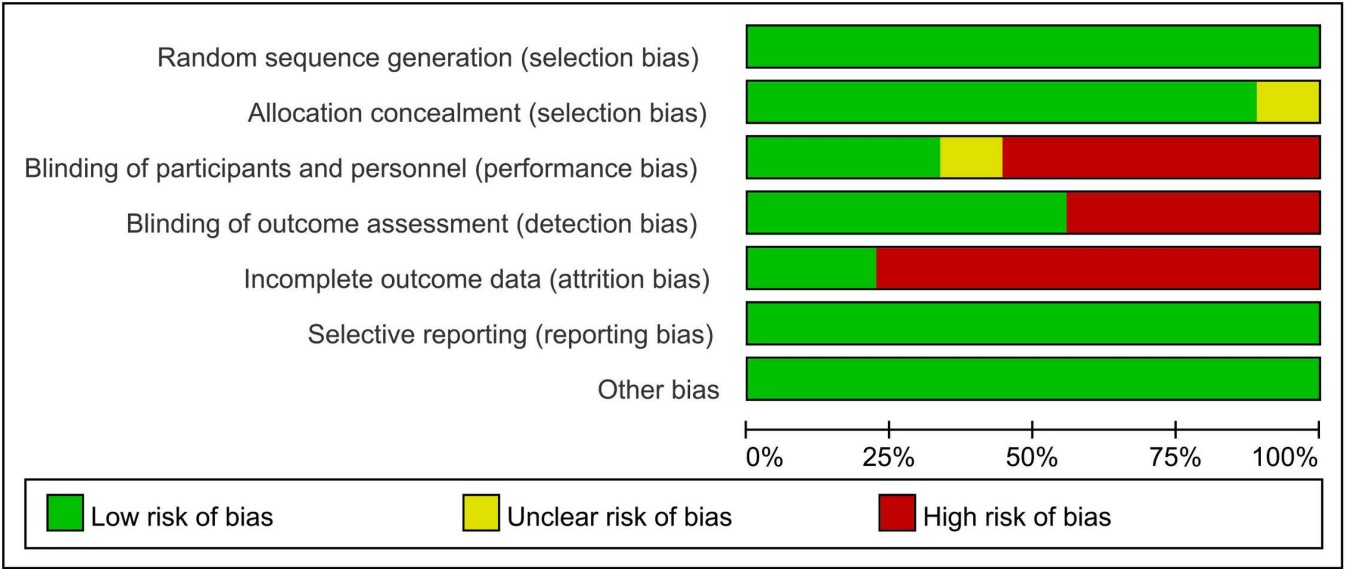

**Fig 2. Risk of bias graph across all included studies.**

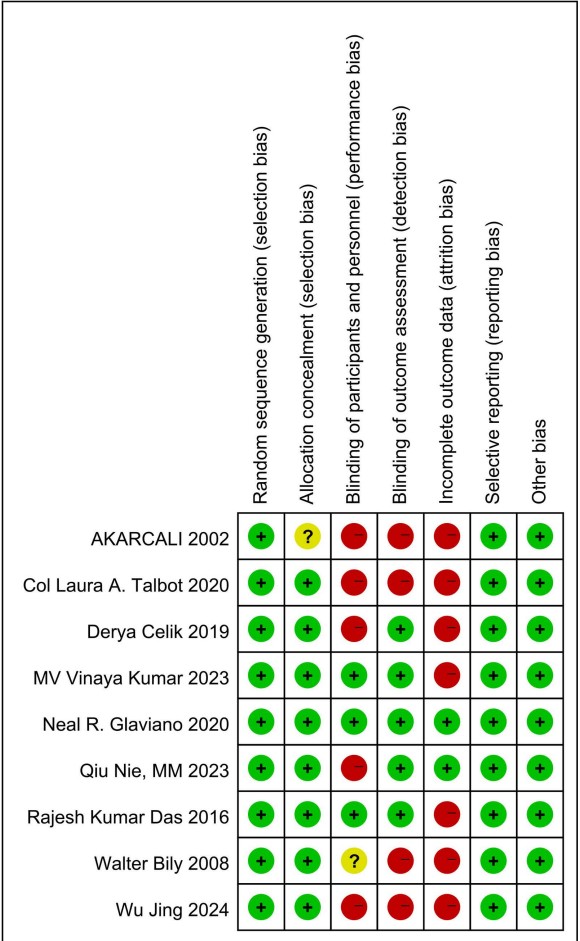

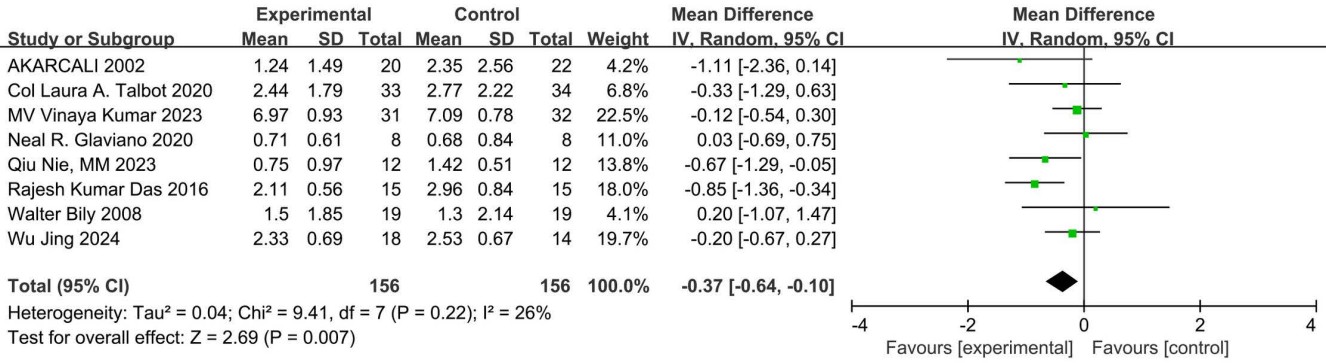

**Fig 3. Risk of bias summary for each included study.**

**Fig 4. Forest plot of pain.**

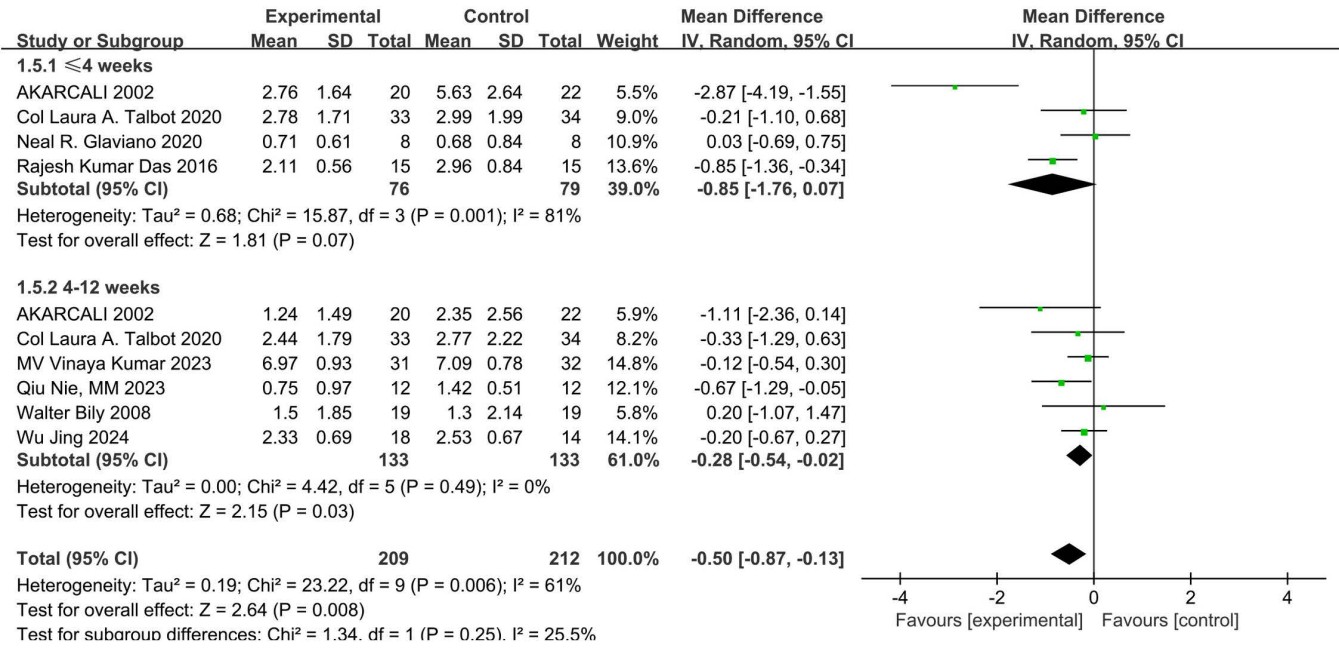

**Fig 5. Forest plot of a subgroup analysis of pain.**

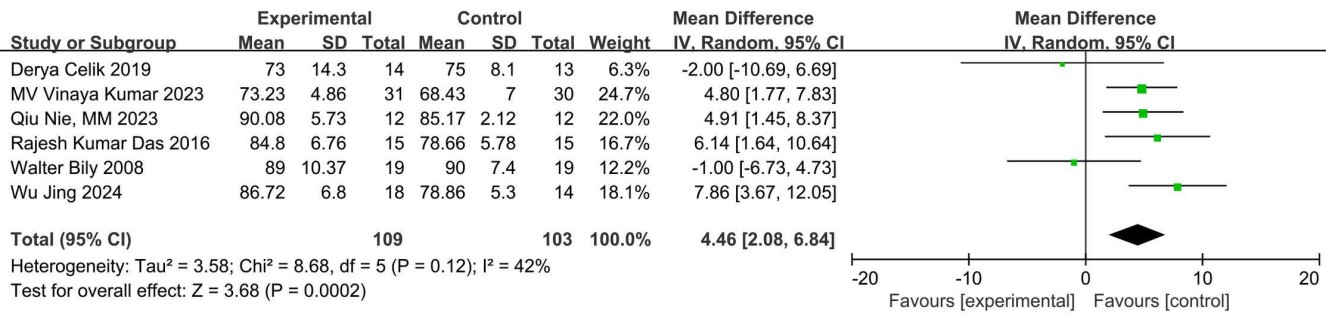

**Fig 6. Forest plot of knee function.**

only (Fig 8). For interventions less than or equal to 4 weeks, the effect was insignificant (SMD: 0.27; 95% CI: −0.24 to 0.78; P = 0.30; low-certainty evidence, $I^2$ = 32%). For 4–12 weeks interventions, NMES combined with exercise therapy significantly increased quadriceps strength compared to exercise alone (SMD: 0.56; 95% CI: 0.16 to 0.96; P = 0.006), with low statistical heterogeneity ($I^2$ = 19%) (Fig 9).

### 4.7. Effect on VMO/VL ratio

Two studies evaluated the effect of NMES on VMO/VL ratio [25,31]. Meta-analysis show that there is very low-certainty evidence and substantial statistical heterogeneity ($I^2$ = 74%) to suggest that adding NMES had no significant impact on the VMO to VL ratio, exhibiting substantial heterogeneity (SMD: 0.8; 95% CI: −0.33 to 1.93; P = 0.16) (Fig 10).

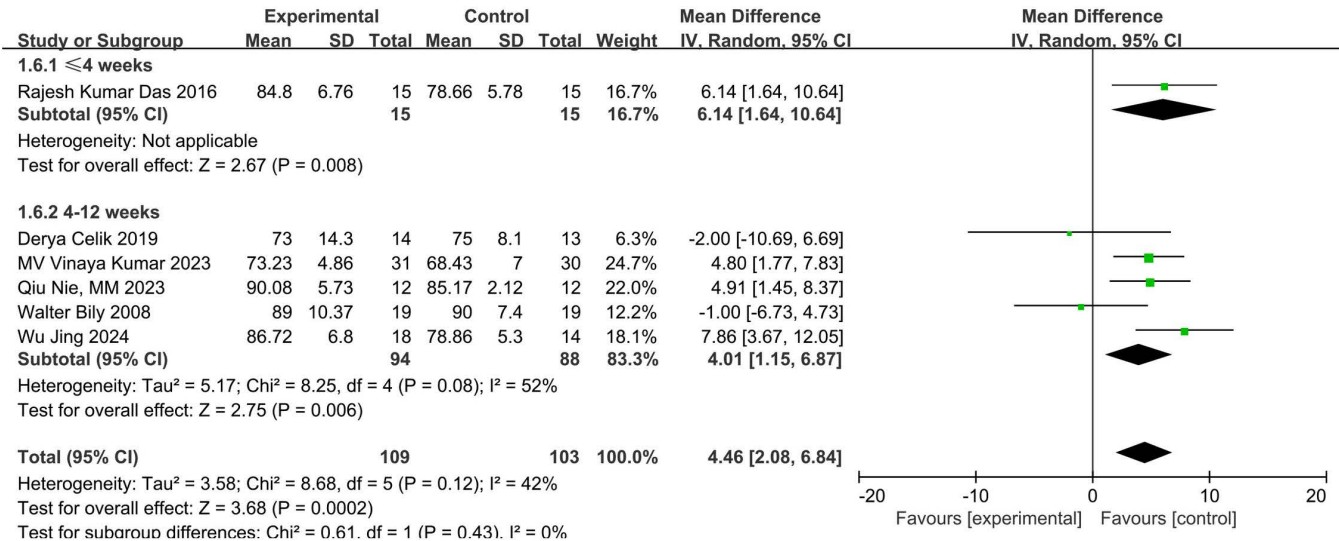

**Fig 7. Forest plot of a subgroup analysis of knee function.**

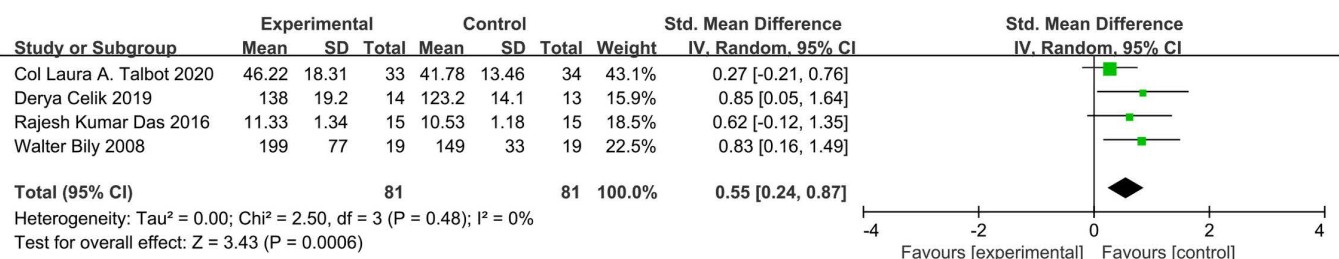

**Fig 8. Forest plot of quadriceps muscle strength.**

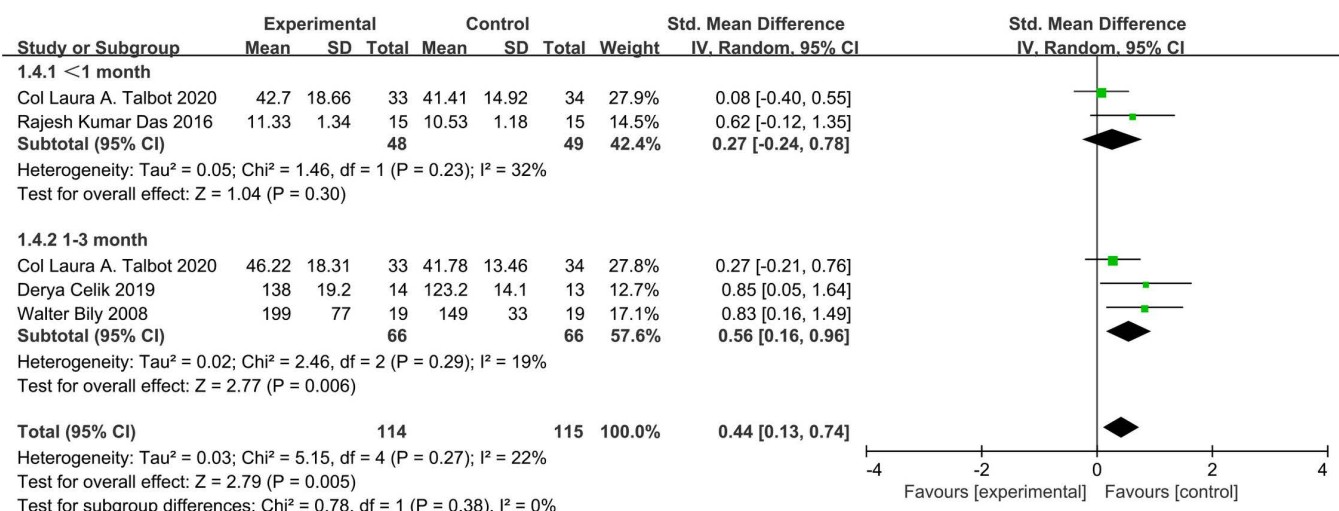

**Fig 9. Forest plot of a subgroup analysis of quadriceps muscle strength.**

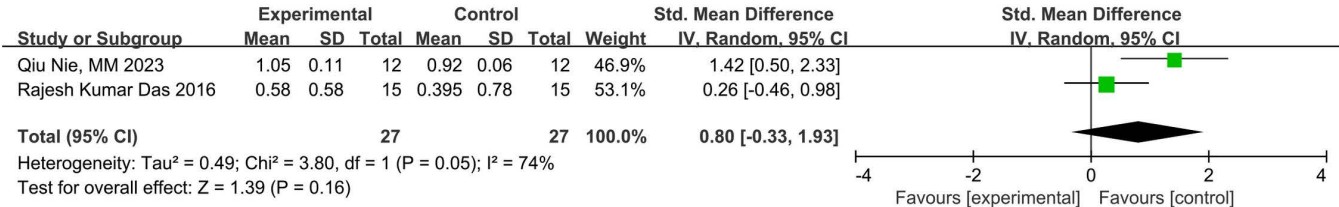

Fig 10. Forest plot of VMO/VL ratio.

## 4.8. Publication bias

Publication bias could not be assessed due to the limited number of trials per outcome.

## 5. Discussion

Previous systematic reviews found that NMES combined with exercise therapy may have additional beneficial effects on pain in PFP patients, but no significant impact on knee function, though these conclusions were based on very low-quality evidence [14]. After expanding the sample size, our study found that compared to exercise therapy alone, NMES combined with exercise therapy may not only further relieve pain but also improve knee function and enhance quadriceps muscle strength, especially when the intervention duration exceeds one month. However, it should be emphasized that these findings are still based on low-quality evidence, and although the level of evidence has slightly improved compared to previous studies, it remains insufficient to draw definitive conclusions.

Pain is a primary symptom in people with PFP, and it can significantly impair knee function, ultimately affecting daily activities [1,9,32]. Our study provides stronger evidence supporting NMES's effectiveness in pain reduction, based on a more comprehensive analysis of high-quality RCTs. The mechanisms by which NMES alleviates pain are complex. NMES induces muscle contraction, which may reduce pain by activating the pain gating theory. This theory posits that pain signal transmission is regulated by spinal cord gating. NMES preferentially stimulates large diameter, myelinated A-α and A-β fibers that transmit non-noxious signals and produce competitive inhibition of pain signals transmitted by smaller A-δ and C fibers. This effectively "closes the gate" on pain transmission, reducing the perception of knee pain. NMES may also reduce pain by promoting vascularization of soft tissues around the patella, improving blood flow, decreasing inflammation, and releasing endogenous analgesic substances [12,33].

Our research found that NMES combined with exercise therapy can further improve knee joint function in PFP patients, which may be related to NMES's ability to relieve pain and increase VMO activation levels [33,34]. Pain reduction enables patients to participate more actively in treatment, enhancing the therapeutic effect of exercise therapy, while increased VMO activation helps improve patellar mechanical trajectory and stability [35]. These synergistic effects collectively enhance the recovery of knee joint function in PFP patients. However, our results are not consistent with previous systematic reviews and some studies [14,23,26]. Possible reasons for this discrepancy may be variations in NMES parameters across studies, such as frequency, duration, and equipment, which could affect treatment outcomes. Given the low-certainty evidence of this result, and the heterogeneity of NMES parameters between included studies, more high-quality RCTs are needed in the future to verify the effect of NMES combined with exercise therapy on knee joint function in PFP patients.

Weak and imbalanced quadriceps strength is a known risk factor for PFP [36,37]. Strengthening the quadriceps helps maintain dynamic knee stability, reduces femoral internal rotation and knee abduction during landing, and minimizes patellar stress [38,39]. Walter Bily et al. concluded that neither exercise therapy alone nor exercise therapy combined with NMES improved quadriceps strength in people with PFP [23]. Prior research indicates that NMES increases muscle

strength through a recruitment pattern different from voluntary movement [40]. While voluntary contractions follow Henneman's Size Principle, where the central nervous system recruits motor units in order from small to large, NMES preferentially activates larger motor units dominated by type II fast-twitch fibers near the electrodes due to their lower resistance [41]. During submaximal exercise therapy, these larger motor units typically remain inactive. When NMES is superimposed on voluntary exercise, it recruits these otherwise inactive larger motor units, potentially explaining the enhanced force production observed with combined therapy [42].

We also assessed the impact of NMES on the VMO/VL activation ratio. Studies have shown that PFP patients may have abnormal VMO/VL activation patterns, which can lead to increased patellar stress [43–47]. However, our analysis, based on two studies, found no significant improvement in the VMO/VL ratio with NMES addition. The heterogeneity of the results and the limited data contribute to low confidence in these findings. More high-quality studies are needed to clarify whether NMES can better balance muscle activation between the VMO and VL.

Furthermore, our subgroup analysis revealed that NMES did not significantly improve pain or quadriceps strength when the treatment duration was less than or equal to 4 weeks. This may be due to the sufficient benefits provided by exercise therapy in the short term, while NMES's effects may be limited in this period [48,49]. Additionally, muscle adaptations typically begin to appear after 4 weeks, but only reach significant enhancement after 8 weeks, therefore, compared to short-term treatment, NMES can demonstrate more pronounced clinical effects only when treatment continues for more than one month [50,51].

This study has several limitations. Many studies experienced participant disengagement, leading to a risk of attrition bias. Subject blinding was also challenging due to the nature of NMES and exercise therapy, introducing potential bias. The level of evidence for the results remains low to very low, requiring higher quality, more rigorously designed studies to verify its clinical effectiveness. The scope of studies was limited, especially regarding the VMO/VL ratio in PFP patients. Furthermore, the NMES parameters used in the included studies varied considerably, contributing to heterogeneity, and the current relevant research has relatively short pulse widths, while studies show that wider pulses are more conducive to motor unit recruitment, which may mask the intervention effect of NMES combined with exercise therapy [52]. Future studies should address these issues, include more high-quality trials, and establish standardized protocols for NMES combined with exercise therapy, to provide more reliable evidence supporting the value of NMES combined with exercise therapy in PFP patients.

Our findings suggest that NMES, when added to the exercise therapy of PFP, can improve quadriceps strength, improve knee function and reduce pain. reduce pain. Given these benefits, clinicians should consider incorporating NMES into treatment plans, especially for patients with quadriceps weakness or dysfunction. It is recommended to combine exercise therapy with NMES for a duration of over 4 weeks to achieve optimal outcomes. NMES can enhance muscle activation and support functional recovery, making it particularly valuable for patients who cannot perform traditional exercises due to pain or muscle inhibition.

## 6. Conclusion

This meta-analysis suggests that combining NMES with exercise therapy provides additional benefits in pain relief and knee function in patients with PFP compared to exercise therapy alone. However, there is minimal evidence suggesting that NMES enhances quadriceps strength or the VMO/VL neuromuscular ratio. To achieve optimal results, it is recommended that NMES treatment last more than 4 weeks. Further high-quality studies are needed to confirm these findings and to standardize treatment protocols.

## Supporting information

**S1 File.  Search strategy.**
(DOCX)

**S2 File. Certainty of evidence.**
(PDF)

**S3 Table. PRISMA checklist.**
(DOCX)

**S4 File. Table of numbering of included studies.**
(DOCX)

**S5 Table. Minimum data set.**
(XLSX)

**S6 File. Risk of bias and quality/certainty assessments.**
(DOCX)

## Acknowledgments

The authors would like to thank Dr. Bin Xu for his assistance in correcting grammatical problems in the manuscript. The authors would like to thank the corresponding authors of the included trials for their assistance in data retrievement.

## Author contributions

**Conceptualization:** Jiawei Zheng.

**Data curation:** Jiawei Zheng, Zixuan Wei, Kunpeng Wang.

**Investigation:** Jiawei Zheng, Zixuan Wei.

**Methodology:** Jiawei Zheng.

**Resources:** Jiawei Zheng.

**Software:** Jiawei Zheng.

**Supervision:** Jiawei Zheng.

**Validation:** Xikai Lin.

**Writing – original draft:** Jiawei Zheng.

**Writing – review & editing:** Huiwu Zuo, Jian Chen.

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
