## [Decision Letter · Decision Letter 0]

Dear Dr. Chen,

Thank you for submitting your manuscript to PLOS ONE. After careful consideration, we feel that it has merit but does not fully meet PLOS ONE’s publication criteria as it currently stands. Therefore, we invite you to submit a revised version of the manuscript that addresses the points raised during the review process.

We look forward to receiving your revised manuscript.

Kind regards,

Tariq Jamal Siddiqi

Academic Editor

PLOS ONE

Journal Requirements:

Please confirm at this time whether or not your submission contains all raw data required to replicate the results of your study. Authors must share the “minimal data set” for their submission. PLOS defines the minimal data set to consist of the data required to replicate all study findings reported in the article, as well as related metadata and methods (https://journals.plos.org/plosone/s/data-availability#loc-minimal-data-set-definition ).

If your submission does not contain these data, please either upload them as Supporting Information files or deposit them to a stable, public repository and provide us with the relevant URLs, DOIs, or accession numbers. For a list of recommended repositories, please see https://journals.plos.org/plosone/s/recommended-repositories .

3. Please amend the manuscript submission data (via Edit Submission) to include author Dr. Jiawei Zhen.

4. Please amend your authorship list in your manuscript file to include author Dr. Jiawei Zheng.

Reviewers' comments:

Reviewer's Responses to Questions

**Comments to the Author**

1. Is the manuscript technically sound, and do the data support the conclusions?

Reviewer #1: Yes

Reviewer #2: Yes

Reviewer #3: Yes

2. Has the statistical analysis been performed appropriately and rigorously?

Reviewer #1: Yes

Reviewer #2: Yes

Reviewer #3: Yes

3. Have the authors made all data underlying the findings in their manuscript fully available?

Reviewer #1: Yes

Reviewer #2: Yes

Reviewer #3: Yes

4. Is the manuscript presented in an intelligible fashion and written in standard English?

Reviewer #1: Yes

Reviewer #2: Yes

Reviewer #3: Yes

Reviewer #1: Zheng et al. conducted a systematic review and meta-analysis on the effects of adding neuromuscular electrical stimulation (NMES) to exercise therapy for patellofemoral pain (PFP). They concluded that NMES reduces pain intensity, improves knee function, and quadriceps strength compared to exercise therapy alone and recommend that interventions should last more than one month to achieve better therapeutic outcomes. However, to improve the manuscript, consider incorporating the following edits.

1. The introduction contains long, convoluted sentences that can be difficult to follow. For example, lines 72-75 "The treatment of PFP is primarily conservative, focusing on rehabilitation through physical therapy and lifestyle modifications, exercise-based rehabilitation is the preferred approach and is advised to be used in conjunction with other supplementary therapies" lacks clarity and transitions poorly. Breaking long sentences into shorter, clearer ones Improves readability and ensures that the text flows more logically, keeping the readers engaged.

2. The introduction contains punctuation errors, structural inconsistencies and abrupt sentence transitions that disrupt the logical flow. For instance lines 90-94 can be revised to "The research further revealed that the combination of NMES with functional training was more effective than treatment in the control group. However, due to conflicting evidence, a systematic evaluation of NMES's role in PFP treatment is necessary." Similarly lines 64-65 can be corrected as “The causes of PFP are multifaceted”. This improves coherence, helping readers follow the logic smoothly, and improves readability of the text.

3. In lines 64-70, the use of consecutive citations like "(5)", "(6)", and "(7)" creates visual clutter and disrupts readability. To improve flow, combine related citations (e.g., "(5-7)") when discussing similar topics. This approach reduces distractions and makes the text more concise and reader-friendly.

4. Lines 75-80: The introduction does not clearly explain why NMES is an effective adjunct for PFP. While it mentions that NMES enhances muscle strength by stimulating motor nerves and causing muscle contractions, it lacks detail on how this mechanism benefits PFP treatment, such as improving VMO activation to address muscle imbalances. The authors should include a concise explanation of how NMES complements conservative treatment by targeting these imbalances, which can alleviate symptoms and improve knee function in PFP patients.

5. The ‘quality assessment’ section of the methods is lengthy and repetitive, making it hard for readers to follow. Phrases like "biases originating from the randomization method" and "biases stemming from variations in planned interventions" can be simplified. Additionally, the structure feels disjointed. The authors should consider revising it to “The revised Cochrane Risk of Bias tool for randomized trials was used by two independent researchers (ZJW and WZX) to assess the methodological quality of the included studies.(18) This tool evaluates potential bias in randomized controlled trials by examining key areas: bias from the randomization process, deviations from intended interventions, missing outcome data, outcome measurement bias, and selective outcome reporting. Each domain was rated as having a low, moderate, or high risk of bias, ensuring a comprehensive assessment of the studies' internal validity.” This improves clarity and creates a smoother flow, making the paragraph more concise and easier to read.

6. The authors in lines 191-194 mention "other techniques" to address heterogeneity, but this phrase is vague and lacks specificity. It's important to clearly state which methods are being used. Providing specific methods helps ensure transparency and allows readers to better understand the robustness of the analysis.

7. The results section is overly detailed, including lengthy explanations of the methodological processes from various studies. The authors should streamline this section by focusing solely on the outcome data and moving the detailed methodological explanations to the discussion. This would improve clarity, make the results easier to interpret, and allow the discussion to focus on contextualizing and analyzing the findings.

8. In lines 278-282, the authors redundantly use the expanded forms of NMES and PFP along with their abbreviations, despite having introduced these earlier in the manuscript. Expanded forms should only be used when first introducing the terms, followed by their abbreviations; thereafter, only abbreviations should be used. This approach ensures consistency, reduces repetition, and improves the overall readability of the manuscript by maintaining a concise and professional tone.

9. The first two paragraphs of the discussion repetitively states that NMES improves pain. This repetition can dilute the impact of the findings and make the discussion less engaging. Summarize key findings without restating them verbatim. Instead, focus on interpreting the results and their implications.

10. In second paragraph of the discussion, while the authors touch on mechanisms by which NMES may alleviate pain (e.g., pain gating theory, increased blood circulation), the explanations are somewhat cursory. Expand on these mechanisms with more detail or cite additional literature to bolster the explanations. Discuss potential biological pathways or neurophysiological mechanisms that could explain the observed effects. This can help clinicians apply NMES more effectively and encourage researchers to explore these pathways further.

11. Third paragraph of the discussion mentions that "the parameters of NMES are different," but it would be more helpful to specify which parameters (e.g., frequency, duration, type of NMES device) have been found to differ and how they specifically affect the results. This would provide clearer context for why the outcomes vary. The second point about the "ceiling effect" is introduced abruptly and lacks elaboration. It would be beneficial to briefly explain what a ceiling effect is in this context and how it relates to the efficacy of NMES in conjunction with exercise therapy.

12. In the second last paragraph of discussion, while the authors rightly suggest the need for more high-quality studies, they do not provide concrete recommendations for what these studies should address. Specify what aspects future studies should investigate, such as standardized NMES protocols, long-term follow-up, or diverse patient populations. Offering clear directions for future research can foster targeted investigations that may resolve current uncertainties in the field.

13. The discussion does not include a dedicated section on clinical implications. It would benefit from a summary that explicitly outlines the potential impact of these findings on clinical practice and patient management. This summary should include specific recommendations for practitioners using NMES in the treatment of PFP. By highlighting the clinical significance of these results, the research can enhance its relevance and encourage practitioners to consider NMES as a valuable adjunct to standard therapeutic approaches.

14. Overall, the manuscript lacks flow and coherence, with certain sections presenting redundant information. Additionally, it contains numerous grammatical, punctuation, and structural inconsistencies. To improve the quality of the manuscript, it is essential to revise for clarity and conciseness, ensuring that each section logically connects to the next. Eliminating redundancy and correcting language errors will enhance readability and strengthen the overall argument. This will not only improve the manuscript's professional presentation but also make the findings more accessible and engaging for readers.

Reviewer #2: The discussion section would benefit from paraphrasing to enhance clarity and ensure a professional tone. Additionally, some sentences contain minor grammatical errors that need correction. I suggest expanding certain points to provide more context and make the narrative more comprehensive. This will strengthen the overall quality and readability of the manuscrip

Reviewer #3: The manuscript still needs proofreading to avoid any grammatical mistakes.

-For Example: In line 65 ("The causes of PFP is multifaceted") weakens the sentence structure. Proper grammar is essential to maintain professionalism.

In Discussion, frequently reiterates similar points about NMES duration and its effect on outcomes, which could make the section overly lengthy and redundant.

The subgroup analysis regarding NMES duration (<1 month vs. ≥1 month) is discussed multiple times, leading to redundancy. The discussion has some inconsistent terminology which needs to be addressed.

**Do you want your identity to be public for this peer review?** For information about this choice, including consent withdrawal, please see our Privacy Policy

Reviewer #1: No

Reviewer #2: **Yes: ** Akash Kumar

Reviewer #3: **Yes: ** Ahmed Kamal Siddiqi

---

## [Author Response · Author response to Decision Letter 1]

20 Jan 2025

Response to reviewers

Dear reviewers:

On behalf of my co-authors, we thank you very much for your letter and comments on our manuscript entitled " The effect of adding neuromuscular electrical stimulation to exercise therapy on patellofemoral pain: A systematic review and meta-analysis" (Manuscript ID: PONE-D-24-36303). We appreciate the editors and reviewers for their constructive and valuable comments. We have revised our manuscript considerably according to the editors’ and reviewers’ comments, questions, and suggestions. In the event that we missed any one of the comments please let us know. This document includes our responses to reviewers and editor comments point by point, and the revised portion are marked in red in our manuscript.

Reply to reviewer 1

Comment 1: The introduction contains long, convoluted sentences that can be difficult to follow. For example, lines 72-75 "The treatment of PFP is primarily conservative, focusing on rehabilitation through physical therapy and lifestyle modifications, exercise-based rehabilitation is the preferred approach and is advised to be used in conjunction with other supplementary therapies" lacks clarity and transitions poorly. Breaking long sentences into shorter, clearer ones Improves readability and ensures that the text flows more logically, keeping the readers engaged.

Reply 1: Thank you for your insightful comments and kind guidance. In the introduction part of the manuscript, we have revised the sentences to improve clarity, readability, and flow. Specifically, we have broken up the longer, more convoluted sentences into shorter, more concise ones to ensure the text flows more logically and is easier for readers to follow. We have rewritten the background part and marked it in red, expecting to meet your requirements.

Comment 2: The introduction contains punctuation errors, structural inconsistencies and abrupt sentence transitions that disrupt the logical flow. For instance lines 90-94 can be revised to "The research further revealed that the combination of NMES with functional training was more effective than treatment in the control group. However, due to conflicting evidence, a systematic evaluation of NMES's role in PFP treatment is necessary." Similarly lines 64-65 can be corrected as “The causes of PFP are multifaceted”. This improves coherence, helping readers follow the logic smoothly, and improves readability of the text.

Reply 2: Thank you for your helpful comments regarding punctuation errors, structural inconsistencies, and abrupt sentence transitions. We have carefully reviewed the manuscript and made the necessary revisions to address these issues. Specifically, we have corrected punctuation, improved sentence structure, and ensured smoother transitions to enhance the logical flow and readability of the text.

Changes in the text:

Page 4, line 65-66 in red: The causes of PFP are multifactorial, with risk factors categorized into extrinsic and intrinsic types.

Page 5, line 92-95 in red: The research further revealed that the combination of NMES with functional training was more effective than treatment in the control group. However, due to conflicting evidence, a systematic evaluation of NMES's role in PFP treatment is necessary

Comment 3: In lines 64-70, the use of consecutive citations like "(5)", "(6)", and "(7)" creates visual clutter and disrupts readability. To improve flow, combine related citations (e.g., "(5-7)") when discussing similar topics. This approach reduces distractions and makes the text more concise and reader-friendly.

Reply 3: Thank you for your insightful feedback regarding the consecutive citations in lines 64-70. We understand how the use of multiple citations in close proximity can create visual clutter and disrupt the flow of the text. In response to your suggestion, we have revised the manuscript by combining related citations where appropriate to improve readability and reduce distractions.

Changes in the text:

Page 4, line 70~71 in red: These factors can lead to patellofemoral malalignment and abnormal patellar movement, resulting in PFP(5-7).

Comment 4: Lines 75-80: The introduction does not clearly explain why NMES is an effective adjunct for PFP. While it mentions that NMES enhances muscle strength by stimulating motor nerves and causing muscle contractions, it lacks detail on how this mechanism benefits PFP treatment, such as improving VMO activation to address muscle imbalances. The authors should include a concise explanation of how NMES complements conservative treatment by targeting these imbalances, which can alleviate symptoms and improve knee function in PFP patients.

Reply 4: Thank you for your careful guidance. We have revised the section to include a more detailed explanation of the mechanism by which NMES helps address muscle imbalances, particularly through its role in improving VMO activation. This, in turn, supports the correction of abnormal patellar tracking and enhances knee function.

Changes in the text:

Page 4, line 78 ~ 83 in red�NMES works by stimulating motor nerves, leading to muscle contractions that strengthen muscles(10, 11). It is commonly used as an adjunct or standalone therapy in knee disorders. In PFP, NMES improves recruitment and activation of the Vastus Medialis Obliquus, helping correct abnormal patellar motion and increasing localized knee compression to reduce pain and improve knee function(12-14).

Comment 5: The ‘quality assessment’ section of the methods is lengthy and repetitive, making it hard for readers to follow. Phrases like "biases originating from the randomization method" and "biases stemming from variations in planned interventions" can be simplified. Additionally, the structure feels disjointed. The authors should consider revising it to “The revised Cochrane Risk of Bias tool for randomized trials was used by two independent researchers (ZJW and WZX) to assess the methodological quality of the included studies.(18) This tool evaluates potential bias in randomized controlled trials by examining key areas: bias from the randomization process, deviations from intended interventions, missing outcome data, outcome measurement bias, and selective outcome reporting. Each domain was rated as having a low, moderate, or high risk of bias, ensuring a comprehensive assessment of the studies' internal validity.” This improves clarity and creates a smoother flow, making the paragraph more concise and easier to read.

Reply 5: Thank you for your helpful suggestions. we have revised this section as you request.

Changes in the text:

Page 8~9, line 164 ~ 171 in red The revised Cochrane Risk of Bias tool for randomized trials was used by two independent researchers (ZJW and WZX) to assess the methodological quality of the included studies(19). This tool evaluates potential bias in randomized controlled trials by examining key areas: bias from the randomization process, deviations from intended interventions, missing outcome data, outcome measurement bias, and selective outcome reporting. Each domain was rated as having a low, moderate, or high risk of bias, ensuring a comprehensive assessment of the studies' internal validity.

Comment 6: The authors in lines 191-194 mention "other techniques" to address heterogeneity, but this phrase is vague and lacks specificity. It's important to clearly state which methods are being used. Providing specific methods helps ensure transparency and allows readers to better understand the robustness of the analysis.

Reply 6: Thank you for your objective suggestions. In Meta-analysis, common solutions to high heterogeneity, in addition to subgroup analysis and sensitivity analysis, primarily involve meta-regression, which uses regression analysis to explore the relationship between study characteristics (such as sample size, intervention intensity, study quality, etc.) and effect size. We have revised this section to make my expression clearer and more specific.

Changes in the text:

Page 10, line 192 ~ 195 in red If the heterogeneity was very large, we performed sensitivity analysis to assess whether the meta-analysis results were robust. In addition, we performed subgroup and meta regression to find sources of heterogeneity.

Comment 7: The results section is overly detailed, including lengthy explanations of the methodological processes from various studies. The authors should streamline this section by focusing solely on the outcome data and moving the detailed methodological explanations to the discussion. This would improve clarity, make the results easier to interpret, and allow the discussion to focus on contextualizing and analyzing the findings.

Reply 7: Thank you for your kind reminder and very constructive comments. We have carefully refined the results, which is actually more consistent with the characteristics of the meta-analysis results. Look forward to your review and suggestions again.

Changes in the text: Page 13 ~ 15, line 244-286

Comment 8: In lines 278-282, the authors redundantly use the expanded forms of NMES and PFP along with their abbreviations, despite having introduced these earlier in the manuscript. Expanded forms should only be used when first introducing the terms, followed by their abbreviations; thereafter, only abbreviations should be used. This approach ensures consistency, reduces repetition, and improves the overall readability of the manuscript by maintaining a concise and professional tone.

Reply 8: Thank you for your thoughtful and friendly reminder. We sincerely apologize for the oversight in lines 278-282, where we inadvertently repeated the expanded forms of NMES and PFP despite having already introduced their abbreviations earlier in the manuscript. We have now corrected this and ensured that only the abbreviations are used following their initial introduction

Changes in the text:

Page 14, line 260 ~ 263 in red The meta-analysis showed that a significant improvement in knee function with NMES (MD:4.75; 95% CI:3.04 to 6.46; p<0.00001; low certainty of evidence) with moderate heterogeneity (I²=42%) (Fig 6).

Comment 9: The first two paragraphs of the discussion repetitively states that NMES improves pain. This repetition can dilute the impact of the findings and make the discussion less engaging. Summarize key findings without restating them verbatim. Instead, focus on interpreting the results and their implications.

Reply 9: Thank you for your objective and constructive suggestions. we have revised the section to avoid redundant statements and have instead focused on summarizing the key findings in a more concise manner. Specifically, we have emphasized the new evidence presented in our study that strengthens the role of NMES in reducing pain levels in PFP, without reiterating previous studies verbatim. We have also highlighted the improved certainty of the evidence from high-quality RCTs, which contrasts with the low-certainty findings of earlier systematic reviews.

Changes in the text:

Page 16 line 295 ~ 300 in red Pain is a primary symptom in individuals with PFP, and it can significantly impair knee function, ultimately affecting daily activities(1, 7, 9). Previous studies have suggested that NMES may reduce pain in PFP, but the evidence in earlier systematic reviews was limited and characterized by low certainty(30, 31). In contrast, our study provides stronger evidence supporting NMES’s effectiveness in pain reduction, based on a more comprehensive analysis of high-quality RCTs.

Comment 10: In second paragraph of the discussion, while the authors touch on mechanisms by which NMES may alleviate pain (e.g., pain gating theory, increased blood circulation), the explanations are somewhat cursory. Expand on these mechanisms with more detail or cite additional literature to bolster the explanations. Discuss potential biological pathways or neurophysiological mechanisms that could explain the observed effects. This can help clinicians apply NMES more effectively and encourage researchers to explore these pathways further.

Reply 10: Thank you for your kind and constructive comments. We appreciate your suggestion to expand on the mechanisms by which NMES alleviates pain, and we have carefully addressed this in our revised manuscript. In response to your comment, we have provided a more detailed explanation of the mechanisms involved in pain relief through NMES.

Changes in the text:

Page 16, line 300 ~ 307 in red The mechanisms by which NMES alleviates pain are complex. NMES induces muscle contraction, which may reduce pain by activating the pain gating theory. This theory posits that pain signal transmission is regulated by spinal cord gating, where NMES stimulates A-α and A-β fibers, which do not transmit pain signals, reducing knee joint pain. NMES may also reduce pain by promoting vascularization of soft tissues around the patella, improving blood flow, decreasing inflammation, and releasing endogenous analgesic substances (10, 32).

Comment 11: Third paragraph of the discussion mentions that "the parameters of NMES are different," but it would be more helpful to specify which parameters (e.g., frequency, duration, type of NMES device) have been found to differ and how they specifically affect the results. This would provide clearer context for why the outcomes vary. The second point about the "ceiling effect" is introduced abruptly and lacks elaboration. It would be beneficial to briefly explain what a ceiling effect is in this context and how it relates to the efficacy of NMES in conjunction with exercise therapy.

Reply 11: Thank you for your constructive suggestions. We have made revisions to this section, where we point out that different frequencies of NMES may lead to varying effects. Additionally, we have provided a brief explanation of the ceiling effect and highlighted its potential impact during the intervention process.

Changes in the text:

Page 16~17, line 311 ~ 320 in red Possible reasons for this discrepancy include variations in NMES parameters such as frequency, duration, and equipment, which can influence treatment outcomes. Michael J. Callaghan et al. (33) demonstrated that variable-frequency NMES more effectively increases quadriceps cross-sectional area than fixed-frequency NMES. This highlights the need for standardized protocols in future studies. Additionally, exercise therapy alone may reach a plateau, known as the ceiling effect, where the nervous system adapts to the NMES stimulus, leading to reduced muscle response over time (34). Therefore, while NMES can be beneficial, its effectiveness may diminish after the initial gains from exercise therapy have been achieved.

Comment 12: In the second last paragraph of discussion, while the authors rightly suggest the need for more high-quality studies, they do not provide concrete recommendations for what these studies should address. Specify what aspects future studies should investigate, such as standardized NMES protocols, long-term follow-up, or diverse patient populations. Offering clear directions for future research can foster targeted investigations that may resolve current uncertainties in the field.

Reply 12: Thank you for your objective and constructive suggestions. We fully agree with the suggestion that providing specific directions for future research will significantly enhance the clarity and impact of our discussion. We have revised this part to include concrete recommendations for future studies.

Changes in the text:

Page 18, line 346 ~ 349�Additionally, the NMES parameters used in the included studies varied considerably, contributing to heterogeneity. The scope of studies was limited, especially regarding the VMO/VL ratio in PFP patients. Future studies should address these issues, include more high-quality trials, and establish standardized protocols for NMES treatment.

Comment 13: The discussion does not include a dedicated section on clinical implications. It would benefit from a summary that explicitly outlines the potential impact of these findings on clinical practice and patient management. This summary should include specific recommendations for practitioners using NMES in the treatment of PFP. By highlighting the clinical significance of these results, the research can enhance its

---

## [Decision Letter · Decision Letter 1]

Dear Dr. Chen,

Thank you for submitting your manuscript to PLOS ONE. After careful consideration, we feel that it has merit but does not fully meet PLOS ONE’s publication criteria as it currently stands. Therefore, we invite you to submit a revised version of the manuscript that addresses the points raised during the review process.

We look forward to receiving your revised manuscript.

Kind regards,

Luciana Labanca

Academic Editor

PLOS ONE

Reviewers' comments:

Reviewer's Responses to Questions

**Comments to the Author**

Reviewer #1: All comments have been addressed

Reviewer #4: (No Response)

Reviewer #5: (No Response)

2. Is the manuscript technically sound, and do the data support the conclusions?

Reviewer #1: Yes

Reviewer #4: No

Reviewer #5: Yes

3. Has the statistical analysis been performed appropriately and rigorously?

Reviewer #1: Yes

Reviewer #4: I Don't Know

Reviewer #5: Yes

4. Have the authors made all data underlying the findings in their manuscript fully available?

Reviewer #1: Yes

Reviewer #4: Yes

Reviewer #5: Yes

5. Is the manuscript presented in an intelligible fashion and written in standard English?

Reviewer #1: Yes

Reviewer #4: No

Reviewer #5: Yes

Reviewer #1: I appreciate the authors for thoroughly addressing the comments. The manuscript has improved significantly; however, a few minor inconsistencies remain that would benefit from further revision. In particular, the Results section contains several structural and grammatical issues that should be carefully reviewed and corrected to enhance clarity and readability.

1. In lines 245-248, the authors should consider replacing the term "superimposed NMES" with "NMES combined with exercise" for clarity and consistency. The term "superimposed" may be ambiguous to some readers, whereas "combined with exercise" more clearly communicates the intervention strategy.

2. The sentence in lines 248-251 is structurally flawed and difficult to follow due to misplaced and redundant elements. Specifically, the phrase "suggest that (I²=81%) indicated..." is grammatically incorrect and confusing. The use of both “suggest” and “indicated” creates unnecessary duplication, and removing one of these words would improve the structure of the sentence and improve comprehensibility.

3. In lines 260–263, the sentence "The meta-analysis showed that a significant improvement in knee function with NMES..." contains a structural error due to the unnecessary use of the word "that." Removing "that" improves grammatical accuracy and sentence flow.

4. In line 281, the sentence "A total of two evaluated the effect of NMES on VMO/VL ratio (17, 26)" would be clearer if revised to include the word "studies." Adding this clarifies the subject of the sentence, improving precision and readability.

Reviewer #4: First, I would like to thank you for the opportunity to review this paper. The authors have made a great effort in putting this piece of work together. However, the manuscript is not ready for publication, and a major revision is necessary. Conducting a systematic review is not an easy task, especially when it is done in a second language. All my comments and suggestions are made solely with the purpose of improving the quality of this systematic review and encouraging the authors to enhance their skills in this type of study.

One general comment I have concerns the readability of the English. Certain parts of the text are difficult to follow, and some terms are used inappropriately (e.g., ‘timing’ instead of ‘time points’). A useful approach to improve this would be to follow the structure of a high-quality systematic review when organising your own. Additionally, consistency in the use of terms and concepts is necessary (e.g., when evaluating the risk of bias). I also suggest changing the term ‘individuals’ to ‘people’.

Introduction

Overall, the introduction is too lengthy. The authors should be more concise, focusing on previously published systematic reviews and clarifying what this review contributes to the existing PFP literature. What distinguishes this review from previous ones?

Line 65-66: The authors state that PFP is a multifactorial condition, with risk factors categorised as extrinsic and intrinsic. However, this categorisation seems unusual, as it is difficult to determine which factors directly cause the onset of PFP symptoms. While it is known that people with PFP may present certain physical characteristics, such as muscle weakness, it is unclear whether these are causative factors for the onset of symptoms. Furthermore, it is not explained how equipment and environment contribute to the onset of PFP symptoms. What are the references supporting these claims? Additionally, how is tension considered an intrinsic risk factor for PFP? I suggest the authors restructure this part of the introduction and provide appropriate references for each statement.

Line 69: The authors appear to place excessive emphasis on muscle delayed activation, which does not seem to be a particularly important factor in people with PFP. As PFP is a multifactorial condition, other factors, such as psychological influences, may also play a significant role.

Line 73: Please provide studies that support the claim that lifestyle changes are important for the treatment of PFP.

Line 75: I suggest changing ‘additional therapies’ to ‘adjunct treatments’.

Lines 76-78: Please provide reference to state that NMES may further enhance recovery.

Lines 84-95: The authors cite three clinical trials in the introduction to support their review. However, it would be more appropriate to reference systematic reviews that have already synthesised this evidence. Existing reviews generally indicate that the effects of using NMES are inconclusive. Citing such reviews would strengthen the introduction and save words.

Recently, Souto et al. published a systematic review evaluating the effectiveness of various adjunct treatments combined with exercise therapy on self-reported pain and functional outcomes. How do the results of the present review differ from those reported by Souto et al.? Has new evidence emerged since that review was published? This should be clarified in the introduction.

Methods

Lines 104-107: Why did the authors not cite the Cochrane Handbook for Systematic Reviews of Interventions? It is a fundamental reference for conducting and reporting systematic reviews.

Lines 118-121: Type of study: The English in this section needs improvement, as it is difficult to follow. Please revise for clarity and coherence.

Lines 123-126: Type of participants: What criteria were used to diagnose PFP? Was it consistent with the 2016 PFP consensus guidelines?

Lines 128-134: Type of interventions: Please review the English in this section for clarity and coherence. Were studies evaluating an NMES sham included?

Line 144: Please specify what is meant by “isometric strength assessments.”

Line 157: Data extraction: Why is the study design being extracted if only RCTs were included?

Lines 163-171: The authors conducted a risk of bias assessment, not a methodological quality or quality assessment. Consistency in terminology is necessary throughout the manuscript.

Line 175: Provide the full name of GRADE (Grading of Recommendations, Assessment, Development and Evaluations) before using the abbreviation.

Lines 186-189: Statistical analysis: Clearly state that I² assesses statistical heterogeneity. A random-effects model should be applied based on methodological heterogeneity, not statistical heterogeneity. How did the authors classify statistical heterogeneity, and what cut-offs were used? I recommend that the authors consult again the Cochrane Handbook for guidance.

Line 197: How was the ‘intervention time’ classified—short, medium, or long? Please clarify by specifying the duration associated with each category.

Results

Line 206: English-language database? However, papers not published in English were also included in these databases. I recommend excluding it.

Line 209: Did the authors attempt to contact the researchers of the unavailable studies? If so, how many attempts were made? Was the standard of three or more attempts followed?

Lines 209-210: Were nine studies included in this meta-analysis or the overall review? Were all nine studies included in the meta-analysis? Please clarify.

Line 216: Please write ‘United States’ instead of ‘US’ to avoid unnecessary abbreviations.

Table 1: Remove the study design, as it is already clear that only RCTs were included. Use IC and CG instead of I and C for improved clarity. Additionally, clarify what QS stands for.

Lines 234-240: Risk of bias and study quality: Ensure consistency in terminology throughout. This paragraph is very unclear — risk of bias should be assessed as ‘low risk’, ‘some concerns’, or ‘high risk’. I recommend mentioning the certainty of the evidence only alongside the pooled results, as you have done elsewhere.

Line 249: The authors mention the level of statistical heterogeneity without having specified it previously in the statistical analysis section. Please provide a clear definition and explanation of how statistical heterogeneity was assessed in the statistical analysis section.

Lines 259-260: Was the abbreviation AKPS introduced earlier in the manuscript? If so, it can be used here; otherwise, provide the full term before using the abbreviation.

In the results section, always clearly specify which intervention was being compared to what. For example:

The meta-analysis showed that a significant improvement in knee function with NMES (MD:4.75; 95% CI:3.04 to 6.46; p<0.00001; low certainty of evidence) with moderate heterogeneity (I²=42%) COMPARED TO EXERCISE ONLY. (Fig 6).

Lines 288-290: Publication bias: You cannot conclude that there is no evidence of bias when each outcome involves fewer than nine trials. In fact, it is more accurate to state that publication bias could not be assessed due to the limited number of trials per outcome.

Discussion

You cannot claim that NMES can effectively enhance pain, function, and strength, as your findings are based on very low to low certainty of evidence.

“Previous studies have suggested that NMES may reduce pain in PFP, but the evidence in earlier systematic reviews was limited and characterized by low certainty.” Your findings do not differ from those reported in previous studies.

Your findings do not differ from those of previous systematic reviews. Additionally, you have not clearly reported the risk of bias for each study. How many studies were assessed as having ‘low risk’?

You should revise your entire discussion, considering that your findings are based on very low to low certainty of evidence. Additionally, avoid using the term ‘treatment’; instead, use ‘exercise therapy’, as that is the focus of your systematic review.

Reviewer #5: GENERAL EVALUATION

The purpose of this study was to investigate the effects of adding neuromuscular electrical stimulation (NMES) to exercise therapy on pain, knee function, quadriceps strength, and the ratio of activation of the Vastus Medialis Obliqus (VMO) to Vastus Lateralis (VL) muscles in people with Patellofemoral pain (PFP). The manuscript’s theme is adequate for publication in PLOS ONE, it is relevant and has novelty for the area of NMES on the rehabilitation of patients with patellofemoral pain. The main findings of the study were that NMES superimposed to exercise therapy reduced pain, improved knee function and increased quadriceps muscle strength when the treatment period was longer than one month compared to exercise therapy alone. The introduction leads the reader to the study problem. The literature reviewed is adequate and updated. However, minor adjustments are needed (Please see Specific Comments). The methods section is relatively clear, but minor changes are needed. The statistical analysis is ok. The results are adequately described and presented in one table and 10 figures. The discussion is clear, and objective and the data support the study conclusion. The references section is adequate. The manuscript was presented in an intelligible fashion and written in Standard English. Minor changes are needed in the manuscript that are described below, and which might help to improve the manuscript’s clarity and quality.

SPECIFIC COMMENTS

TITLE: Ok.

ABSTRACT

Lines 37-38: The number of participants is not correct: 171+166=337 and not 360. Please correct.

INTRODUCTION

Line 67. To which equipment are you referring to? Physical fitness equipment, equipment used at work?

Line 68. While strength and force are usually related to muscle function, tension is usually used in bone, ligament, tendon and cartilage function, as they passively generate tension to support the forces produced by muscle contraction or external forces applied to the human body. Perhaps specifying which tension you are referring to here will help the reader to better understand the intrinsic risk factors for the development of PFP.

Lines 78-79. NMES stimulates not motor nerves, but motor and sensory neurons, as most nerves are composed of both sensory and motor neurons. Discomfort is mostly a result from the activation of sensory receptors at the skin, tendon and ligament receptors and activation of sensory neurons by NMES. But here you need to change the word “nerve” by “neurons”.

METHODS

Lines 120-121. The sentence is missing something here, such as “were applied”.

Line 139: Please correct the name of the muscle: Vastus Medialis Oblique.

Line 144: Please specify what you mean by other isometric strength assessments.

Line 156. Name of the leading author.

STATISTICAL ANALYSIS

Ok.

RESULTS

Line 206. In the abstract you refer to five databases and here you refer to four databases. Which is the correct one?

Lines 214-215. Like the comment made for the abstract, the number of participants is not correct: 171+166=337 and not 360. Please correct.

Lines 217-218. I think that you are missing two studies here regarding the duration of the interventions.

Table 1. I would reorganize the studies in the table alphabetically by the authors’ last names, as this makes it easier for the reader to find the studies. In addition, you need to use only the last name of the first author with the name with the first letter capitalized (e.g., Akarcali et al., 2013; Bily et al., 2008; Celik et al., 2019; Das et al., 2016; Glaviano et al., 2020; Jing et al., 2024; Kumar et al., 2023; Mv et al., 2023; Nie et al., 2024; Talbot et al., 2020).

Please, also correct the abbreviation NMES in the table. I also suggest that all the abbreviations are presented alphabetically, as this helps the reader to find the description of each abbreviation easily.

Line 270. You mention six studies, but you show only five within the parentheses.

Line 281. “Two studies evaluated the effect…”

DISCUSSION

Lines 302-304. It is not clear in your explanation regarding the pain gating theory why by stimulating A-α and A-β nerve fibers, NMES reduces the pain transmitted by pain receptors. You also need to specify which nerve fibers do transmit pain signals so that the reader understands the difference between nerve fibers that transmit and those that do not transmit pain stimuli.

Lines 316-318. This sentence is confusing as you start referring to the fact that exercise therapy alone may reach a plateau or ceiling effect, and then you continue saying that the nervous system adapts to the NMES stimulus. Is the ceiling effect related only to exercise therapy or to both exercise therapy and NMES? As you started this paragraph calling the reader’s attention to the fact that your study found that NMES contributes to improved knee function, it appears to me that you are discrediting your results while contrasting them to the existent literature. Which one should the reader follow: NMES really contributes to improving knee function or does NMES not really contribute to knee function?

Line 324. I think that you are using here PT as an abbreviation for Physical Therapy. However, in previous paragraphs you used physical therapy, not the abbreviation. It is also not clear to me why you use in most of the manuscript the expression exercise therapy and not physical therapy. I would perhaps use the term physical therapy in the beginning (with the abbreviation PT after the first mention) and make it clear that you are referring to studies that used exercises during physical therapy. On the other hand, if you want to maintain the term exercise therapy, then perhaps do not use physical therapy in any part of the manuscript.

Lines 327-331. You refer here to prior research, but it is not clear if you are referring only to reference 38. In addition, you may need to better describe how NMES superimposed to voluntary exercises may increase muscle force production. While during voluntary exercises the central nervous system recruits motor units through Henneman’s Size Principle from the smaller to the larger motor units, NMES recruits motor units that are closer to the electrodes placed on the skin, but with a preferential recruitment of larger motor units (constituted of type II fast-twitch muscle fibers) due to the lower resistance of the large diameter of their axons compared to the higher resistance of the smaller (type I slow-twitch muscle fibers) motor units’ axons. In other words, during exercise therapy larger motor units are probably not recruited due to the submaximal level of the exercises, and superimposing NMES will lead to the recruitment of large motor units that are not recruited during the voluntary exercises.

Another aspect that you may add regarding the exercise therapy time and the NMES effects is that usually muscle adaptations occur after four weeks but with higher increases occurring after eight weeks of treatment. Therefore, 8-12 weeks will show a higher clinical effect compared to short-term (i.e., up to 4 weeks) exercise therapy.

Finally, you may want to read manuscripts about wide-pulse NMES (i.e., with a larger pulse duration of 1 ms per pulse phase). This NMES type also recruits motor units via reflex pathways, thereby increasing the number of motor units recruited, which will generate a higher force production. In all the studies you showed in the review, pulse duration was short (65-400 μs) and is a limitation, and if you want to improve the NMES effects on muscle strength superimposed on voluntary exercises, new electrical stimulators with larger pulse duration also need to be introduced in clinical practice.

CONCLUSION

Ok.

REFERENCES

Please review the references as I think that some are missing some data. I am also not sure about the format of the journals’ names, but I usually use all journal names with the first letter of each word capitalized.

GENERAL COMMENT

I would add a space between the parentheses of the references and the words preceding the parentheses in all the text.

**Do you want your identity to be public for this peer review?** For information about this choice, including consent withdrawal, please see our Privacy Policy

Reviewer #1: No

Reviewer #4: No

Reviewer #5: **Yes: ** MARCO AURELIO VAZ

---

## [Author Response · Author response to Decision Letter 2]

2 May 2025

Response to reviewers

Dear reviewers:

On behalf of my co-authors, we thank you very much for your letter and comments on our manuscript entitled " The effect of adding neuromuscular electrical stimulation to exercise therapy on patellofemoral pain: A systematic review and meta-analysis" (Manuscript ID: PONE-D-24-36303R1). We appreciate the editors and reviewers for their constructive and valuable comments. We have revised our manuscript considerably according to the editors’ and reviewers’ comments, questions, and suggestions. In the event that we missed any one of the comments please let us know. This document includes our responses to reviewers and editor comments point by point, and the revised portion are marked in red in our manuscript.

Reply to reviewer #1

Comment 1: I appreciate the authors for thoroughly addressing the comments. The manuscript has improved significantly; however, a few minor inconsistencies remain that would benefit from further revision. In particular, the Results section contains several structural and grammatical issues that should be carefully reviewed and corrected to enhance clarity and readability.

Reply 1: We sincerely thank the reviewer for their constructive feedback and encouraging assessment of our revised manuscript. We are pleased to hear that our previous revisions have significantly improved the paper. We acknowledge the reviewer's observation regarding minor inconsistencies in the Results section. We have carefully reviewed this section again and corrected all issues to enhance clarity and readability. We appreciate the reviewer's attention to detail, which has helped us further refine the quality of our manuscript

Comment 2: In lines 245-248, the authors should consider replacing the term "superimposed NMES" with "NMES combined with exercise" for clarity and consistency. The term "superimposed" may be ambiguous to some readers, whereas "combined with exercise" more clearly communicates the intervention strategy.

Reply 2: Thank you for your careful guidance. We agree that the term "superimposed NMES" may be ambiguous to some readers. As per your recommendation, we have replaced this term with "NMES combined with exercise" throughout the manuscript to improve clarity and maintain consistency in describing our intervention strategy.

Page 13, line 231 in red: NMES combined with exercise significantly reduced pain compared to controls

Comment 3: The sentence in lines 248-251 is structurally flawed and difficult to follow due to misplaced and redundant elements. Specifically, the phrase "suggest that (I²=81%) indicated..." is grammatically incorrect and confusing. The use of both “suggest” and “indicated” creates unnecessary duplication, and removing one of these words would improve the structure of the sentence and improve comprehensibility.

Reply 3: We appreciate the reviewer's careful assessment of our manuscript. We agree that the sentence in lines 248-251 contained redundant elements with both 'suggest that' and 'indicated' creating an awkward structure. We have revised the sentence to eliminate this duplication, improving clarity and readability.

Page 13, line 232-235 in red: In subgroup analyses, very low-certainty evidence and considerable statistical heterogeneity (I²=81%) suggested no significant effect when the intervention was less than or equal to 4 weeks (MD: -0.85; 95% CI: -1.76 to 0.07; P=0.07) (Fig 5).

Comment 3: In lines 260–263, the sentence "The meta-analysis showed that a significant improvement in knee function with NMES..." contains a structural error due to the unnecessary use of the word "that." Removing "that" improves grammatical accuracy and sentence flow.

Reply 3: Thank you for identifying this grammatical oversight. We have revised the sentence in lines 260–263 by removing the redundant "that," as recommended.

Page 14, line 245-248 in red: The meta-analysis showed a significant improvement in knee function with NMES (MD:4.46; 95% CI:2.08 to 6.84; p=0.0002; low certainty of evidence) with moderate statistical heterogeneity (I²=42%) compared to exercise only (Fig 6).

Comment 4: In line 281, the sentence "A total of two evaluated the effect of NMES on VMO/VL ratio (17, 26)" would be clearer if revised to include the word "studies." Adding this clarifies the subject of the sentence, improving precision and readability.

Reply 4: Thank you very much for your insightful feedback on improving the clarity of our manuscript. We sincerely appreciate your attention to detail and the constructive suggestions provided. We have revised the sentence in line 281 to include the word "studies" for clarity.

Page 15, line 269 in red: Two studies evaluated the effect of NMES on VMO/VL ratio.

We sincerely thank the reviewers for their thorough evaluation and valuable feedback during this second round of review. Your insightful comments and constructive suggestions have significantly enhanced the quality and rigor of our meta-analysis. We deeply appreciate the time and expertise you have contributed to improving our work, and we look forward to any further recommendations you may have to further refine this study.

Reply to reviewer #4

Comment 1: First, I would like to thank you for the opportunity to review this paper. The authors have made a great effort in putting this piece of work together. However, the manuscript is not ready for publication, and a major revision is necessary. Conducting a systematic review is not an easy task, especially when it is done in a second language. All my comments and suggestions are made solely with the purpose of improving the quality of this systematic review and encouraging the authors to enhance their skills in this type of study.

Reply 1: Thank you for your valuable review of our manuscript. We deeply appreciate the time and expertise you have dedicated to evaluating our work. Your constructive feedback is invaluable, and we are genuinely grateful for your thorough assessment.

Your comments have helped us recognize the significant shortcomings in our systematic review methodology and writing. Your insights have highlighted areas where our approach needed substantial improvement. We have carefully reviewed all your suggestions and have made comprehensive revisions throughout the manuscript accordingly. Your guidance has been instrumental in enhancing not only this particular paper but also our understanding of systematic review methodology more broadly.

Comment 2: One general comment I have concerns the readability of the English. Certain parts of the text are difficult to follow, and some terms are used inappropriately (e.g., ‘timing’ instead of ‘time points’). A useful approach to improve this would be to follow the structure of a high-quality systematic review when organising your own. Additionally, consistency in the use of terms and concepts is necessary (e.g., when evaluating the risk of bias). I also suggest changing the term ‘individuals’ to ‘people’.

Reply 2: Thank you sincerely for your careful review and feedback on our manuscript's overall writing and readability. After carefully consulting multiple high-quality systematic reviews, we have made diligent efforts to thoroughly revise the entire text, including the use of relevant terminology and concepts, to improve the clarity and quality of our paper.

Comment 3: Overall, the introduction is too lengthy. The authors should be more concise, focusing on previously published systematic reviews and clarifying what this review contributes to the existing PFP literature. What distinguishes this review from previous ones?

Reply 3: We sincerely thank the reviewer for this constructive feedback regarding our introduction. We have thoroughly revised it to make it more concise and coherent. Regarding the contribution of this review to the existing PFP literature and what distinguishes it from previous reviews, we have written a separate paragraph in the introduction to clearly state this.

Comment 4: Line 65-66: The authors state that PFP is a multifactorial condition, with risk factors categorised as extrinsic and intrinsic. However, this categorisation seems unusual, as it is difficult to determine which factors directly cause the onset of PFP symptoms. While it is known that people with PFP may present certain physical characteristics, such as muscle weakness, it is unclear whether these are causative factors for the onset of symptoms. Furthermore, it is not explained how equipment and environment contribute to the onset of PFP symptoms. What are the references supporting these claims? Additionally, how is tension considered an intrinsic risk factor for PFP? I suggest the authors restructure this part of the introduction and provide appropriate references for each statement.

Reply 4: We sincerely thank the reviewer for this insightful observation regarding our categorization of risk factors for PFP. After reviewing multiple relevant studies, we realized that categorizing PFP risk factors as intrinsic and extrinsic factors is not very accurate. We have now revised this section, referencing the statement from the International Patellofemoral Pain Research Retreat, to categorize them as distal, proximal, and local factors, with appropriate references provided for each factor. Regarding the term "tension," we apologize for not being clear. Here we are referring to the tension of the quadriceps and hamstring muscles, as research has shown that excessive tightness in these muscles is one of the risk factors for PFP.

Page 3-4, line 60-69 in red: Prospective research has identified that patellofemoral pain etiology can be categorized into proximal, local, and distal factors [2]. Proximal factors involve hip muscle weakness leading to altered lower limb kinematics during weight-bearing activities [3]. Local factors include quadriceps dysfunction [4], altered Vastus Medialis Oblique (VMO)/ Vastus Lateralis (VL) activation timing [5], quadriceps and hamstring tightness [6]. Distal factors encompass foot and ankle mechanics, such as increased navicular drop, which affect tibial rotation and subsequently alter patellofemoral joint mechanics [7]. All these factors contribute to PFP by creating abnormal patellofemoral joint mechanics and increased joint stress.

Comment 5: Line 69: The authors appear to place excessive emphasis on muscle delayed activation, which does not seem to be a particularly important factor in people with PFP. As PFP is a multifactorial condition, other factors, such as psychological influences, may also play a significant role.

Reply 5: We sincerely thank the reviewer for this insightful comment. We agree that our original text may have overemphasized this single factor without adequately acknowledging the complex, multifactorial nature of PFP. We have revised it to "altered VMO/ VL activation timing" and avoided suggesting it as the primary or most important factor.

Page 3-4, line 63-65 in red: Local factors include quadriceps dysfunction [4], altered Vastus Medialis Oblique (VMO)/ Vastus Lateralis (VL) activation timing [5], quadriceps and hamstring tightness [6].

Comment 6: Line 73: Please provide studies that support the claim that lifestyle changes are important for the treatment of PFP.

Reply 6: We appreciate the reviewer's careful assessment of our manuscript. After reviewing relevant literature, we believe that lifestyle changes can improve symptoms in PFP patients to some extent, but it is not one of the important and preferred treatment methods for PFP. Therefore, we have removed this expression from the manuscript.

Page 4, line 70-73 in red: The primary treatment for PFP is conservative, focusing on rehabilitation through exercise therapy, which is widely recognized as the main treatment for musculoskeletal conditions including PFP and is often combined with adjunct treatments to achieve optimal clinical outcomes [8, 9].

Comment 7: Line 75: I suggest changing ‘additional therapies’ to ‘adjunct treatments’.

Reply 7: Thank you for this helpful suggestion regarding terminology. We have made the corresponding changes in the manuscript.

Page 4, line 72-73 in red: often combined with adjunct treatments to achieve optimal clinical outcomes [8, 9].

Comment 8: Lines 76-78: Please provide reference to state that NMES may further enhance recovery.

Reply 8: Thank you for your valuable feedback regarding of our manuscript. We acknowledge the need to provide proper references supporting our statement that NMES may further enhance recovery. We have added the relevant references in the manuscript to support this viewpoint.

Page 4, line 73-76 in red: Neural muscular electrical stimulation (NMES), as a painless and non-invasive technique, is typically used as an adjunct treatment method to further promote recovery [10, 11].

10. Hauger AV, Reiman MP, Bjordal JM, Sheets C, Ledbetter L, Goode AP. Neuromuscular electrical stimulation is effective in strengthening the quadriceps muscle after anterior cruciate ligament surgery. Knee surgery, sports traumatology, arthroscopy : official journal of the ESSKA. 2018;26(2):399-410. Epub 2017/08/19. doi: 10.1007/s00167-017-4669-5. PubMed PMID: 28819679.

11. Borzuola R, Laudani L, Labanca L, Macaluso A. Superimposing neuromuscular electrical stimulation onto voluntary contractions to improve muscle strength and mass: A systematic review. European journal of sport science. 2023;23(8):1547-59. Epub 2022/07/21. doi: 10.1080/17461391.2022.2104656. PubMed PMID: 35856620.

Comment 9: Lines 84-95: The authors cite three clinical trials in the introduction to support their review. However, it would be more appropriate to reference systematic reviews that have already synthesised this evidence. Existing reviews generally indicate that the effects of using NMES are inconclusive. Citing such reviews would strengthen the introduction and save words.

Reply 9: Thank you for this valuable feedback. According to the feedback, we have revised this section, removed citations of individual clinical trials and instead citing high-quality systematic reviews to indicate that whether NMES combined with exercise therapy can produce better clinical benefits for PFP patients remains uncertain.

Page 4-5, line 77-87 in red: However, whether NMES combined with exercise therapy can produce better clinical benefits for PFP patients currently has no definitive conclusion. A recent systematic review indicated that NMES combined with exercise therapy can slightly improve pain in patients with PFP, but does not significantly improve knee function. However, the evidence certainty of these results is very low[14]. Furthermore, the review did not evaluate key indicators such as muscle strength and lacked subgroup analysis based on intervention duration. Given the emergence of more evidence and the weakness of existing evidence, further systematic reviews are necessary to clarify the clinical value of NMES as an adjunct treatment for patients with PFP.

Comment 10: Recently, Souto et al. published a systematic review evaluating the effectiveness of various adjunct treatments combined with exercise therapy on self-reported pain and functional outcomes. How do the results of the present review differ from those reported by Souto et al.? Has new evidence emerged since that review was published? This should be clarified in the introduction.

Reply 10: We appreciate the reviewer's thoughtful comment. Our system evaluation results differ from those reported by Souto et al., with specific details discussed in the paper's discussion section. Additionally, compared to Souto et al.'s system evaluation, we included more literature and added two new outcome measures: quadriceps strength and VMO/VL activation ratio. We also conducted subgroup analysis based on intervention time to determine the impact of different intervention durations on PFP patients.

Comment 11: Why did the authors not cite the Cochrane Handbook for Systematic Reviews of Interventions? It is a fundamental reference for conducting and reporting systematic reviews.

Reply 11: Thank you for your valuable feedback regarding our citation practices. We acknowledge this oversight in our manuscript. The Cochrane Handbook is recognized as a gold standard resource that provides comprehensive methodolo

---

## [Editor Report · Decision Letter 2]

The effect of adding neuromuscular electrical stimulation to exercise therapy on patellofemoral pain�A systematic review and meta-analysis

PONE-D-24-36303R2

Dear Dr. Chen,

We’re pleased to inform you that your manuscript has been judged scientifically suitable for publication and will be formally accepted for publication once it meets all outstanding technical requirements.

Kind regards,

Luciana Labanca

Academic Editor

PLOS ONE
---

## [Editor Report · Acceptance letter]

PONE-D-24-36303R2

PLOS ONE

Dear Dr. Chen,

I'm pleased to inform you that your manuscript has been deemed suitable for publication in PLOS ONE. Congratulations! Your manuscript is now being handed over to our production team.

Kind regards,

on behalf of

Dr. Luciana Labanca

Academic Editor

PLOS ONE